# The Effect of a Pulmonary Rehabilitation on Lung Function and Exercise Capacity in Patients with Burn: A Prospective Randomized Single-Blind Study

**DOI:** 10.3390/jcm9072250

**Published:** 2020-07-15

**Authors:** Yu Hui Won, Yoon Soo Cho, So Young Joo, Cheong Hoon Seo

**Affiliations:** 1Department of Physical Medicine and Rehabilitation, Research Institute of Clinical Medicine of Jeonbuk National University–Biomedical Research Institute of Jeonbuk National University Hospital, Jeonju 54907 Korea; wonyh@jbnu.ac.kr; 2Department of Rehabilitation Medicine, Hangang Sacred Heart Hospital, College of Medicine, Hallym University, 94-200 Yeongdeungpo-Dong Yeongdeungpo-Ku, Seoul 07247, Korea; hamays@hanmail.net (Y.S.C.); anyany98@naver.com (S.Y.J.)

**Keywords:** inhalation burn, pulmonary rehabilitation, pulmonary function test

## Abstract

We performed pulmonary function (PF) tests and factors affecting PF evaluation in 120 patients with inhalation injury to evaluate the effects of pulmonary rehabilitation (PR) in burn patients with inhalation injury. Patients were randomized into pulmonary rehabilitation (PR) group and conventional rehabilitation (CON) group. PF tests, including forced vital capacity (FVC), 1-s forced expiratory volume FEV1), maximum voluntary ventilation (MVV), and respiratory muscles strength (maximal expiratory pressure (MEP) and maximal inspiratory pressure (MIP)), were measured by mouth pressure meter in the sitting position. Diffusing capacity for carbon monoxide (DLco) was determined by the single-breath carbon monoxide technique. Peak cough flow (PCF) was measured by a peak flow meter. Diaphragmatic mobility (DM) was evaluated on anteroposterior fluoroscopy. All evaluations were performed in all groups at baseline and after 12 weeks. There were no differences in evaluations between the PR group and CON group before the intervention. There were significant improvements in the PCF and MIP (%) changes, taken before and after rehabilitation in the PR group, compared with the changes in the CON group (*p* = 0.01, and *p* = 0.04). There were no significant changes in the other parameters in the PR group compared with the changes in the CON group (*p* > 0.05). There were significant differences in DLco (%), MIP, MIP (%), and DM between the PR group and the CON group (*p* = 0.02, *p* = 0.005, and *p* = 0.001) after 12 weeks of rehabilitation. There were no differences between the PR group and CON group after 12 weeks rehabilitation in the other parameters (*p* > 0.05). PR for patients with major burns and smoke inhalation induced improved PCF, MIP, MIP (%), DLco (%), and DM. These results show that PR should be a fundamental component of the treatment program for patients with burns.

## 1. Introduction

Pulmonary complications occur in 15–25% of hospitalized patients with large surface burn and inhalation [1]. Pulmonary dysfunction can be developed as a result of complications caused by direct thermal injury to the respiratory tract, smoke irritations, and respiratory tract infection. Although the survival rates of patients with burn injury have increased because of improved acute management, pulmonary complications have continued to be the main cause of mortality in patients with burns [2,3,4]. Mlcak et al. reported that the children with major burns decreased pulmonary function (PF) up to 8 years after injuries [5]. The patients who survive suffer from persistent physiological impairments, limited aerobic capacity, increased skeletal muscle catabolism, and decreased muscle strength/endurance [6,7,8]. One of the causes of persistent physiological impairments is a decrease in PF [7]. Rehabilitation strategies for the long-term recovery in patients with burns aim to improve the physical function and independence in daily activities [9]. PR using aerobic exercise and resistive exercise was effective even in the chronic period of two years after burn injury for aerobic capacity [10].

Some studies have reported successful increases in PF and aerobic capacity after 12 weeks of a pulmonary rehabilitation (PR) program using aerobic exercise on treadmill and ergometer on chronic obstructive pulmonary disease (COPD) patients [11]. Additionally, PR programs using individually targeted exercises provide opportunities to improve PF, exercise tolerance, dyspnea symptoms, and skeletal muscles dysfunction in patients with chronic respiratory disorders (such as asthma and interstitial lung disease) [12]. Recent studies with thermal injury have shown that resistive exercises combined with aerobic exercises improve PF and muscle strength of the lower extremities [13,14,15]. Ozkal et al. provides a strong suggestion for a personalized PR, including respiratory muscle strengthening [8]. Although the effect on resistive exercise and aerobic exercises was progressed in burn patients with reduced PF, randomized controlled trial on the role of PR known to be effective is needed.

Acute management of patients with inhalation injury include deep breathing exercises, therapeutic coughing, chest physiotherapy, and early ambulation [3,16]. Standard treatment protocols beyond the acute period for pulmonary dysfunction in patients with inhalation injury have not been established. We evaluated PF and factors affecting PF in patients with burns to evaluate the effects of PR, including resistance exercise, aerobic exercise, and deep breathing exercise.

## 2. Experimental Section

### 2.1. Methods

#### Study Design and Statement of Ethics

This was a single-blinded, randomized controlled trial. Between October 2019 and January 2020, participants who have been admitted to the Department of Rehabilitation Medicine for the first time after the acute phase treatment were recruited at Hangang Sacred Heart Hospital, Seoul, Korea. The trial was registered at ClinicalTrials.gov (NCT04125108). After explaining the purpose of this study and the side effects (such as muscle soreness and dyspnea during rehabilitation), the participants decided whether or not to participate. Participants were provided with the written informed consents.

### 2.2. Study Group

A major burn is defined as a burn covering 25% or more of total body surface area, and burns that involve the face, hands, feet, genitalia, and perineum [17]. The inclusion criteria were as follows: Between 18 and 75 years of age; partial or full-thickness major burns that healed spontaneously or required skin grafting; requirement of assisted ventilator care during intensive care unit (ICU) management, and diagnosis of smoke inhalation based on the history of smoke exposure and fibroptic bronchoscopy. Exclusion criteria were as follows: Current smokers, who had concomitant intrinsic lung diseases, patients who could not hold breaths due to vocal cord palsy; patients who underwent intubation or tracheostomy at the time of evaluation; patients with full or virtually full-thickness involvement of at least one-half of the total thorax, patients who underwent an escharectomy on the thorax, and patients who took medications (such as bronchodilators) that affect PF.

### 2.3. Intervention

All participants received standard treatment on hypertrophic scars, which involved pain medication, scar lubrication, burn scars massage therapy, and occupational therapy. The occupational therapy focused on burned upper extremity treatment, such as passive range of motion exercise, daily activity living training, and manual lymphatic drainage in the upper extremities.

PR programs consisted of circuit training. The exercise circuit consists of the following stations: Four resistive exercises, aerobic exercise, and deep breathing exercises. The four basic resistance exercises included four sets of lower limb exercises (leg press and knee extension), and upper limb exercises (chest press and biceps curl). All exercises were performed using variable resistance machines. The resistance exercise apparatus consists of five stages, and it is practiced at 50–80% (8–12 repetitions) from the level corresponding to the maximum strength [18]. Load intensity was adjusted every four weeks according to maximum strength test. PR programs also included aerobic conditioning exercises on the treadmill. The intensity in aerobic training was to be performed on the basis of the subjective sensation of dyspnea of the participants, measured the Borg scale. When the participants reported a dyspnea sensation with values between 4 and 6 on the Borg scale, the intensity was maintained [19]. Resistance exercise 30 min per session and aerobic exercise 30 min per session were conducted five days per week, with lasting one hour daily. The patients were instructed to perform deep breathing exercises. Verbal instructions were given during aerobic and resistance exercises, and deep expiration and deep inspiration were performed without forcing abdominal retraction. The CON group participated in a conventional physical rehabilitation, including a range of motion exercises and ambulatory training. Conventional rehabilitation included overground aerobic exercise training, where the patients went around the therapeutic room track. When the participants reported a dyspnea sensation with values between 4 and 6 on the Borg scale, the intensity was maintained. Conventional therapy was performed for one hour daily. The exercise frequency and duration did not differ between the PR group and CON group. All exercise sessions were supervised by a trained physiotherapist.

### 2.4. Outcome Measures

The spirometry study variables included forced vital capacity (FVC) and 1-s forced expiratory volume (FEV1). We calculated predicted values based on age, weight, and height. Relative values were reported as FVC (%) and FEV1 (%) [20]. Lung parenchymal injury was measured by diffusing capacity for carbon monoxide (DLco) [17]. DLco was determined by the single-breath carbon monoxide technique. DLco changes can be substantial as a function of hemoglobin (Hb) level [21]. Hb levels were evaluated during respiratory examination. The FVC, FEV1, and DLco were measured using a Quark PFT (Cosmed, Italy). Maximal respiratory pressure and reflecting muscles strength were measured by a mouth pressure meter (Pony FX; COSMED, Rome, Italy) in the sitting position. The highest maximal expiratory pressure (MEP) and maximal inspiratory pressure (MIP) value in three or more attempts were chosen. We calculated the predicted MEP and MIP values based on age, height, and weight [22]. Peak cough flow (PCF) was measured using a peak flow meter. For PCF measurement, the patients were asked to cough as much as possible through a peak flow meter. The highest value in each parameter was obtained by conducting at least three trials. Before the measurement of diaphragmatic mobility (DM) using fluoroscopy, the diaphragm was trained by performing diaphragmatic breathing exercises. Experiments were conducted with the patients lying on a radioscopic table in the supine position. DM was measured by viewing the diaphragm movement displayed on the fluoroscopy device. The images of maximum expiration and inspiration were recorded on the same film (Figure 1). Three measurements were performed for each patient, and the best value obtained was recorded. All evaluations were assessed at baseline and after 12 weeks of intervention in the respiratory laboratory. When appropriate, the evaluations were used according to the guidelines established by the American Thoracic Society [23].

### 2.5. Statistical Analysis

Statistical analysis was performed using SPSS version 23 (IBM Corp., Armonk, NY, USA). The parametric measurements between the PR group and CON group were analyzed using an independent t-test after the normality test. The non-parametric measurements between the two groups were analyzed using the Mann-Whitney test. To examine the pretreatment homogeneity between the two groups, the Mann-Whitney test was used for TBSA, age, postburn day studied, ICU days, and ventilator days with a significance level of *p* < 0.05. To examine the pretreatment homogeneity between the PR group and CON group, the Fisher exact test was used for sex and burn types, and the Pearson Chi-square test was used for tracheostomy history and the presence of chest wall burns. The changes before and after intervention were compared using the Mann-Whitney test for all parameters between groups. The parameters after intervention were compared between the two groups using the independent t-test for the MEP (%) and DLCO. Other parameters except for the MEP (%) and DLCO were analyzed using the Mann-Whitney test. A significant difference was considered when the *p*-value was <0.05.

## 3. Results

This study enrolled 120 patients who were diagnosed with inhalation injury and major burns. Numbers were assigned according to the order of admissions. A computer program was used to randomly divide them into the PR group (*n* = 60) and the conventional (CON) group (*n* = 60). Moreover, two patients in the PR group completed more than 45 sessions before dropping out, due to scheduling conflict. Two patients in the CON group were forced to drop out before completing the study, including one who completed 30 sessions before dropping out (due to unrelated medical issues) and one who completed 50 sessions before dropping out (due to a personal scheduling conflict). Two and four patients from the CON group and PR group, respectively, were excluded because the patients had not wanted outpatient follow-up due to no symptoms after discharge (Figure 2).

There were no differences between the PR group and CON group before the interventions with respect to the baseline characteristics, such as sex, age, and burn type. Moreover, the factors that could influence PF (such as TBSA, inhalation history, length of ICU stay, ventilator days, chest wall burn and tracheostomy history) were similarly distributed in two groups (Table 1). No differences in PF, DLco, respiratory muscle strengths, DM, and PCF at baseline were found in the PR group and CON group at baseline (Table 2).

There were significant improvements in the PCF changes taken before and after rehabilitation in the PR group compared with the changes in the CON group (*p* = 0.01) (Table 3). There were no significant changes in the FVC (%), FVC, FEV1 (%), FEV1, and MVV in the PR group compared with the changes in the CON group (*p* = 0.23, *p* = 0.32, *p* = 0.61, *p* = 0.54, and *p* = 0.33). There were no significant changes in the DLco (%) and DLco in the PR group compared with the changes in the CON group (*p* = 0.47 and *p* = 0.96). There were significant improvements in the changes of the MIP (%) taken before and after rehabilitation in the PR group compared with the changes in the CON group (*p* = 0.04). There were no significant changes between two groups in MIP, MEP, and MEP (%) (*p* = 0.73, *p* = 0.12, and *p* = 0.13). There were no significant changes between the two groups in DM (*p* = 0.61).

There were significant differences in the DLco (%), MIP, MIP (%), and DM between the PR group and the CON group (*p* = 0.02, *p* = 0.005, *p* = 0.001, and *p* = 0.005) after 12 weeks of rehabilitation (Table 4). There were no differences between the PR group and CON group after 12 weeks rehabilitation in the other parameters except for the DLco (%), MIP, MIP (%), and DM (*p* > 0.05).

## 4. Discussion

Our results indicate that there were improvements in the PCF, MIP, MIP (%), DLco (%), and DM in the PR group after 12 weeks of PR compared with the parameters in the CON group. This is the first randomized controlled trial on the use of the PR program, including resistive exercise, aerobic exercise, and deep breathing exercise, to recover PF and factors affecting PF (such as DLco, DM, and respiratory muscle strength) after thermal injury in adult patients.

PR that has been proven so far is known for resistance training, aerobic exercise, and respiratory muscle training [11,12,24]. Vogiatzis et al. [11] reported improvements in the physiologic functions in patients with COPD after aerobic exercises. Alison et al. reported improvements in PF measures (FEV1, FVC, MIP, and MEP) after PR programs using respiratory muscle resistance training [24]. Inspiratory muscle training improved inspiratory muscle strength, and increased weak DM after cardiac surgery [25]. DM is associated with exercise intolerance, and Yamaguti et al. have emphasized the importance of deep breathing exercises [26]. PR protocols have not been established in patients with decreased PF [27]. In this study, a PR using resistive exercise, aerobic exercise, and deep breathing exercise, which is known to be effective in COPD, was applied to the patients with inhalation injury.

PF studies, DLco, DM, and respiratory muscles strengths were used as evaluation tools to confirm the effects of the PR program. Smoke inhalation causes increases in a ventilation-perfusion mismatch and airway obstruction. After the major burn has healed, the scars restricted the thorax, and the pulmonary reserve was lost [28]. Airflow obstruction degrees measured by FVC, FEV1, and FEV1/FVC%, restrictive disorder degrees measuring by TLC, and reduced DLco were observed in patients with inhalation injury [29]. The patients subsequently demonstrate a reduction in the DLco, FVC, and TLC for as long as several years after the injury [5,29]. A significant correlation was observed between chest imaging and PF (FVC, FEV1, TLC, and DLco). Change in chest imaging correlated well with the change in the FVC [17]. Peak expiratory flow changes with time in patients with inhalation injury represented the course of the disease [30]. In addition to PF and DLco, respiratory muscles strengths and DM were additionally evaluated to confirm the factors that may cause respiratory improvement in this study.

There were significant improvements in the changes of the MIP (%) and PCF taken before and after rehabilitation in the PR group compared with the changes in the CON group. Coughing has a function to clear airway secretions. The MEP and MIP showed a significant correlation with PCF [31]. MEP is crucial for an effective cough. The lung volume attained before coughing by inspiratory muscles is also important. The inspiratory muscle strength also affects deep breathing. Decreased function of deep breathing worsens chest wall rigidity and burdens on weakened inspiratory muscles [32]. The patients with respiratory muscles weakness cannot take deep breaths, which is needed to maintain the lung capacity [33]. In this study, it is thought that deep breathing exercise in PR program caused improvement of inspiratory muscles strength and PCF. PF is affected by decreases in both MIP and MEP [32,34]. Physical impairment should be affected by inspiratory and expiratory muscle strengths [35,36]. Coughing performance and only inspiratory muscles strengths improved, so there were no significant changes in PF. These results are consistent with other studies which the exercise capacity improved, but PF in the burned patient decreased for many years [37]. After 12 weeks of PR, there were significant differences in the DLco (%), MIP, MIP (%), and DM between groups. A strong correlation was found between the DM, PF (VC), and the function of deep breathing in patients with COPD [38]. The significance of physiologic tests for measuring DLco is that they permit the diagnosis of an impaired surface area for the transfer of gases [17]. Significantly low values of lung diffusing capacity in patients with smoke inhalation injury confirm the inhalation injury has an impact on the proof of gas exchange [29]. Deep breathing exercises are more effective in improving the ventilatory function in patients with inhalation injury than incentive spirometry exercises [39]. The improvement of ventilatory function was confirmed by significant differences of Dlco, and the similar results to the previous study were confirmed. The PR program, including resistive exercise, aerobic exercise, and deep breathing exercise, has been proven to be effective on patients with acute burn injuries.

For a better understanding of the pathophysiology of pulmonary dysfunction on patients with burns and inhalation injury, further studies comparing PF between patients with burns and age-matched healthy subjects are required. Additionally, further study is required on the extended pulmonary rehabilitation period and variety in the intensities of exercise. There is a study that the 6-min walking test (6MWT) is related to PF in burn patients [8], but the 6MWT was not routinely performed because there were patients with lower extremity burns among the inhaled burn patients. It is thought that the measurement of the total lung capacity (TLC) is required when evaluating the correct PF. Since there is no equipment capable of measuring the TLC, it was replaced by the DLco test.

## 5. Conclusions

Our results demonstrate that the patients with large burn surface and inhalation injury can improve their PCF, MIP, DLco, and DM by a PR program, which should be an essential component of a multidisciplinary treatment program for patients with thermal injury. Furthermore, pulmonary rehabilitation is feasible, inexpensive, and promising for application in physiotherapy.

## Figures and Tables

**Figure 1 jcm-09-02250-f001:**
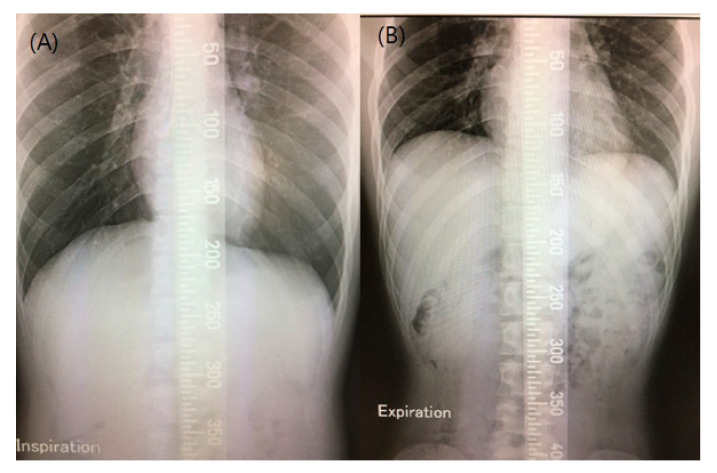
Measurements of diaphragm mobility. Chest radiography in anteroposterior view during maximal inspiration and maximal expiration conducted on the same film. (**A**) Chest X-ray taken during a maximal inspiratory maneuver. (**B**) Chest X-ray taken during a maximal expiratory maneuver.

**Figure 2 jcm-09-02250-f002:**
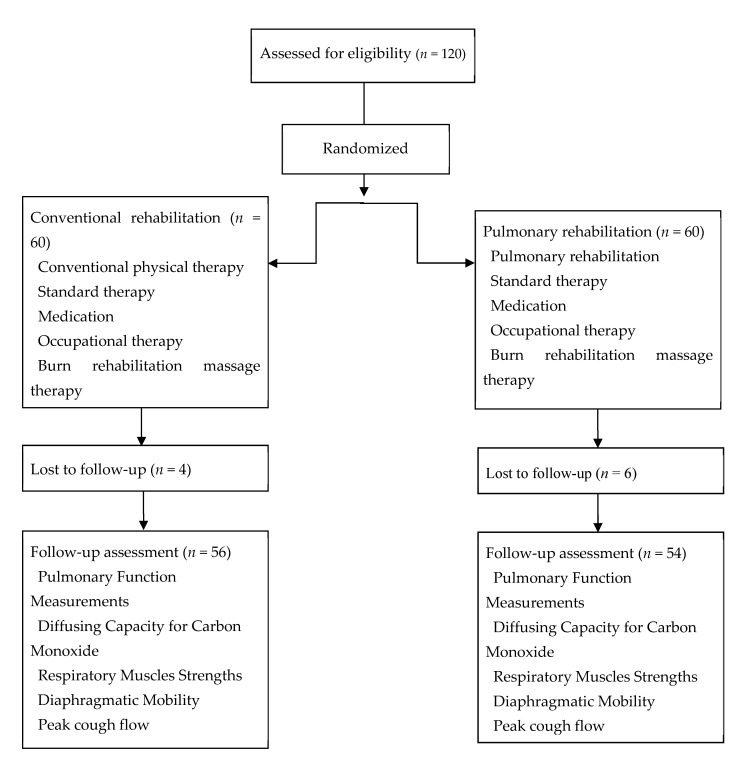
Diagram for subject enrollment, allocation and follow up.

**Table 1 jcm-09-02250-t001:** Demographic and Clinical Characteristics of Subjects.

Variables	PR Group (*n* = 54)	CON Group (*n* = 56)	*p*
Male:Female, *n*	52:2	56:0	0.24
Age (years)	43.59 ± 13.07	46.82 ± 13.91	0.23
TBSA (%)	36.93 ± 21.18	40.29 ± 19.18	0.36
Postburn Day Studied	85.81 ± 34.37	90.39 ± 38.76	0.65
Type of burn, *n* FB:EB:CoB:CeB	52:0:0:2	48:2:2:4	0.19
ICU days	27.30 ± 16.25	25.39 ± 22.93	0.20
Ventilator days	9.81 ± 11.74	10.43 ± 14.12	0.91
Tracheostomy history, *n*	18	16	0.47
Chest wall burn, *n* (anterior and circumferential)	20	12	0.06
Hb	13.94 ± 2.63	13.04 ± 1.43	0.08

PR = pulmonary rehabilitation; CON = conventional rehabilitation; TBSA = total burn surface area; FB = flame burn; EB = electrical burn; CoB = contact burn, CeB = chemical burn; Hb = hemoglobin; Values are presented as mean ± standard deviation.

**Table 2 jcm-09-02250-t002:** Pulmonary function measurements at baseline.

Variables	PR Group (*n* = 54)	CON Group (*n* = 56)	*p*
FVC (%)	90.07 ± 12.21	90.89 ± 15.84	0.28
FVC (mL)	3.89 ± 0.69	3.99 ± 0.75	0.45
FEV1 (%)	86.93 ± 17.45	88.93 ± 15.47	0.60
FEV1(mL)	3.04 ± 0.68	3.14 ± 0.64	0.42
MVV (L/min)	95.42 ± 30.57	94.39 ± 23.43	0.29
DLco (mL/min/mmHg)	19.06 ± 4.71	19.48 ± 3.66	0.61
DLco (%)	81.30 ± 13.03	76.93 ± 17.67	0.06
PCF (L/min)	417.78 ± 121.45	442.50 ± 96.54	0.31
MIP (cmH_2_O)	89.00 ± 27.17	85.55 ± 23.82	0.50
MIP (%)	89.44 ± 24.24	85.82 ± 19.79	0.55
MEP (cmH_2_O)	97.44 ± 27.60	90.33 ± 17.51	0.16
MEP (%)	71.78 ± 17.36	66.66 ± 13.36	0.30
DM	5.32 ± 1.22	4.75 ± 1.44	0.17

PR = pulmonary rehabilitation; CON = conventional rehabilitation; FVC, forced vital capacity; FEV1, 1-s forced expiratory volume; FEV1/FVC ratio; MVV, Maximum voluntary ventilation; DLco, diffusing capacity for carbon monoxide; PCF, peak cough flow, MIP, maximum inspiratory pressure; MEP, maximum expiratory pressure; DM, diaphragmatic mobility; Values are presented as mean ± standard deviation.

**Table 3 jcm-09-02250-t003:** The changes of pre-intervention and post-intervention in both groups.

Variables	PR Group (*n* = 54)	CON Group (*n* = 56)	*p*
FVC (%)	3.41 ± 4.50	3.43 ± 10.73	0.23
FVC (mL)	0.12 ± 0.16	0.15 ± 0.49	0.32
FEV1 (%)	0.85 ± 5.11	2.46 ± 10.42	0.61
FEV1 (mL)	0.06 ± 0.29	0.10 ± 0.38	0.54
MVV (L/min)	4.10 ± 21.54	9.29 ± 19.75	0.33
DLco (mL/min/mmHg)	1.48 ± 2.64	1.55 ± 1.86	0.96
DLco (%)	6.44 ± 10.98	6.18 ± 8.02	0.47
PCF (L/min)	52.41 ± 113.00	6.21 ± 103.76	* 0.01
MIP (cmH_2_O)	18.26 ± 18.68	11.84 ± 20.57	0.73
MIP (%)	17.04 ± 18.93	10.07 ± 19.97	* 0.04
MEP (cmH_2_O)	2.63 ± 15.75	7.32 ± 26.52	0.12
MEP (%)	1.70 ± 11.87	4.05 ± 16.68	0.13
Diaphragmatic mobility	0.50 ± 1.11	0.50 ± 1.41	0.61

PR = pulmonary rehabilitation; CON = conventional rehabilitation; FVC, forced vital capacity; FEV1, 1 s forced expiratory volume; FEV1/FVC ratio; FEF 25–75, forced expiratory flow rate between 25 and 75% of the FVC; PEF, peak expiratory flow; MVV, Maximum voluntary ventilation; DLco, diffusion capacity for carbon monoxide; PCF, peak cough flow, MIP, maximum inspiratory pressure; MEP, maximum expiratory pressure; DM, diaphragmatic mobility; Values are mean ± standard deviation., * *p* < 0.05 Mann-Whitney test, the changes of pre-intervention and post-intervention were compared between groups.

**Table 4 jcm-09-02250-t004:** Pulmonary function measurements after 12 weeks.

Variables	PR Group (*n* = 54)	CON Group (*n* = 56)	*p*
FVC (%)	93.48 ± 12.11	94.32 ± 11.25	0.57
FVC (mL)	4.01 ± 0.75	4.14 ± 0.64	0.50
FEV1 (%)	87.78 ± 17.12	91.39 ± 12.44	0.40
FEV1 (mL)	3.10 ± 0.68	3.25 ± 0.59	0.35
MVV (L/min)	99.52 ± 30.96	103.65 ± 22.08	0.49
DLco (mL/min/mmHg)	20.54 ± 4.86	21.03 ± 3.69	0.56
DLco (%)	87.74 ± 14.81	83.11 ± 15.95	* 0.02
PCF (L/min)	470.19 ± 136.02	448.71 ± 91.92	0.16
MIP (cmH_2_O)	107.26 ± 23.21	97.39 ± 25.36	* 0.005
MIP (%)	106.48 ± 21.08	70.71 ± 20.61	* 0.001
MEP (cmH_2_O)	100.07 ± 28.03	96.04 ± 26.37	0.30
MEP (%)	73.48 ± 16.70	70.71 ± 20.61	0.44
Diaphragmatic mobility	5.83 ± 1.21	5.25 ± 1.16	* 0.005

PR = pulmonary rehabilitation; CON = conventional rehabilitation; FVC, forced vital capacity; FEV1, 1 s forced expiratory volume; FEV1/FVC ratio; FEF 25–75, forced expiratory flow rate between 25 and 75% of the FVC; PEF, peak expiratory flow; MVV, Maximum voluntary ventilation; DLCO, diffusion study; PCF, peak cough flow, MIP, maximum inspiratory pressure; MEP, maximum expiratory pressure, Values are presented as mean ± standard deviation. * *p* < 0.05

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
