# Peer review of "The Effect of a Pulmonary Rehabilitation on Lung Function and Exercise Capacity in Patients with Burn: A Prospective Randomized Single-Blind Study"

_jcm, 2020, doi:10.3390/jcm9072250_

Round 1

Reviewer 1 Report

The authors have conducted a RCT to evaluate the role of pulmonary rehabilitation in 120 patients with major burns, finding an improvement in PCF, MIP, DLCO and DM at 12 weeks.

Major comments:

  • A proper evaluation of the DLCO should include correction for hemoglobin levels. It is mandatory that at least mean Hb values corresponding to the time of DLCO evaluation (baseline and 12 weeks) are included both in the table legends and in the text.
  • Were pulmonary function tests and DLCO performed with the same machine? This need to be specified.
  • As the authors properly discuss, TLC is an important outcome measure in the follow-up of patients with acute lung injury due to smoke inhalation. Is it possible to show TLC values? Otherwise, the reason why it was not tested should be disclosed.
  • Similarly, why was 6MWT not performed at baseline?
  • The authors state tracheostomy was an exclusion criterion; however, they report comparability at baseline for tracheostomy history, which applies to a high number of patients (18 vs. 16). This needs further elucidation.
  • A table reporting major comorbidities and smoking history must be included.

Minor comments:

  • Was this a single-center study? This needs to be clarified in the text.
  • The authors should clarify which are the "medications that affect PF" they have considered as an exclusion criterion.
  • Why was Student's t-test used only for MEP (%) and DLCO?
  • Moderate English or formal changes are required (e.g. lines 62, 64, 75, 97, 101, 106, 160-161, 279).
  • Figure 1 has a limited interest in its current state. Please provide a more explicative figure legend or remove it. 

Author Response

The authors have conducted a RCT to evaluate the role of pulmonary rehabilitation in 120 patients with major burns, finding an improvement in PCF, MIP, DLCO and DM at 12 weeks.

1. A proper evaluation of the DLCO should include correction for hemoglobin levels. It is mandatory that at least mean Hb values corresponding to the time of DLCO evaluation (baseline and 12 weeks) are included both in the table legends and in the text. Were pulmonary function tests and DLCO performed with the same machine? This need to be specified.The FVC, FEV1, and DLco were measured using a Quark PFT (Cosmed, Italy). We added the descriptions of outcome measures in the method section.Answer> We appreciate you careful advise. The hemoglobin (Hb) correction in the DLCO measurement is mandatory. We added the mean Hb values in the Table 1, and added the reference to DLco evaluation in the method section.

2. As the authors properly discuss, TLC is an important outcome measure in the follow-up of patients with acute lung injury due to smoke inhalation. Is it possible to show TLC values? Otherwise, the reason why it was not tested should be disclosed. Similarly, why was 6MWT not performed at baseline? Answer> We agree with the reviewer. We added the limitation that the 6MWT could not be performed due to burns in the lower extremities. It is thought that the measurement of the TLC is required when evaluating the correct PF. We added to the limit that the institution was replaced by the DLco because there was no equipment to measure the TLC.

3. The authors state tracheostomy was an exclusion criterion; however, they report comparability at baseline for tracheostomy history, which applies to a high number of patients (18 vs. 16). This needs further elucidation. 

Answer> We appreciate you careful advise. We will recruit and analyze more experimental groups in the future for patients undergoing airway incision.

4. A table reporting major comorbidities and smoking history must be included.

Answer> We appreciate you careful advise. We added the exclusion criterias (current smokers).

5. Was this a single-center study? This needs to be clarified in the text. 

Answer> We appreciate you careful advise. This study was conduced in a single-center study. We added the name of the institution where the study was conduced in the method section.

6. The authors should clarify which are the "medications that affect PF" they have considered as an exclusion criterion. 

Answer> We agree with the reviewer. A description for the medications was added in the method section.

7. Why was Student's t-test used only for MEP (%) and DLCO?

Answer> We appreciate you careful advise. Statistical method was described in detail for each item. We hope this will help the reader to understand with ease.

8. Moderate English or formal changes are required (e.g. lines 62, 64, 75, 97, 101, 106, 160-161, 279). 

Answer> We appreciate you careful advise. English proofreading was performed throughout the paper.

9. Figure 1 has a limited interest in its current state. Please provide a more explicative figure legend or remove it. 

Answer> We appreciate you careful advise. We remove the Figure 1. We hope this will help the reader to understand with ease.

Reviewer 2 Report

Overall this is a very good study with good recruitment. There is important information and I would like to see it published but it does need extra information and the discussion needs totally rewriting. 

Method 

Inclusion criteria was "requirement for assisted ventilator care during intensive care" but Exclusion criteria was "patients who underwent intubation". What do you mean by assisted ventilator care, was this only non-invasive ventilation? 

If the patients were ventilated please specify period of ventilation. I find it doubtful that they were in intensive care for an average of 27 days but not ventilated.

"Major burns" is a bit vague what % was the exclusion criteria?

Did any patients have thoracic surface burns that required an escharotomy if so this should be analysed separately or acknowledged. 

Were the measurements done in a pulmonary function laboratory?

As 120/120 patients were recruited did they have a chance to refuse? This is why I query ethics. 

The figure of 50-80% for resistance work is very variable. If it is based on the patient's initial effort why wasn't this standard?

Discussion

The discussion needs totally rewriting . It is very disjointed, with poor English, does not flow at all and does not really attempt to explain why the improvements were seen. It requires a patho-physiological description of the damage that inhalation injury does and how this was supposed to have have been improved by the exercises. The discussion currently refers mainly to pulmonary diseases. 

 Author Response

Overall this is a very good study with good recruitment. There is important information and I would like to see it published but it does need extra information and the discussion needs totally rewriting. 

  1. Inclusion criteria was "requirement for assisted ventilator care during intensive care" but Exclusion criteria was "patients who underwent intubation". What do you mean by assisted ventilator care, was this only non-invasive ventilation? Answer> We appreciate you careful advise. We added more detailed descriptions of the participants in the method section. We added the period of ventilation in the Table 1. We hope this will help the reader to understand with ease. If the patients were ventilated please specify period of ventilation. I find it doubtful that they were in intensive care for an average of 27 days but not ventilated.

    Answer> We appreciate you careful advise. We added more detailed descriptions of the participants in the method section. We added the period of ventilation in the Table 1. We hope this will help the reader to understand with ease.

2. "Major burns" is a bit vague what % was the exclusion criteria?

Answer> We agree with the reviewer. We added the definition of major burns in the method section.

3. Did any patients have thoracic surface burns that required an escharotomy if so this should be analysed separately or acknowledged. 

Answer> We appreciate you careful advise. In this study, we evaluate to confirm the effects of pulmonary rehabilitation in patients with inhalation and major burns. We excluded the patients with full or virtually full-thickness involvement of at least one-half of the total thorax, and patients who underwent an escharectomy on the thorax.

4.Were the measurements done in a pulmonary function laboratory?

Answer> We appreciate you careful advise. We added the descriptions of outcome measure in the method section.

5. As 120/120 patients were recruited did they have a chance to refuse? This is why I query ethics. 

Answer> We agree with the reviewer. We added a process for obtaining consent in order to fit the research ethics of patients in the method section.

6. The figure of 50-80% for resistance work is very variable. If it is based on the patient's initial effort why wasn't this standard?

Answer> We agree with the reviewer. The contents of the method section were added with more references.

7. The discussion needs totally rewriting . It is very disjointed, with poor English, does not flow at all and does not really attempt to explain why the improvements were seen. It requires a patho-physiological description of the damage that inhalation injury does and how this was supposed to have have been improved by the exercises. The discussion currently refers mainly to pulmonary diseases.
Answer> We appreciate you careful advise. The contents of the discussion section were rewritten as you pointed out. We hope this will help the reader to understand with ease.

Round 2

Reviewer 1 Report

In their revised version of this manuscript, the authors have incorporated all the changes I required. I have no further comments.

Reviewer 2 Report

I am satisfied that all the necessary changes have been made. 

The English style could still be improved.